# Barriers and facilitators to implementing six monthly multi-month dispensing of antiretroviral therapy in two urban HIV clinics during the COVID-19 Era in Malawi

Rachel Chamanga[1,2]*, Agatha Bula[1,3], Denview Magalasi[4], Stella Mahuva[5], Mulinda Nyirenda[6,7], Kwasi Torpey[8,9], Thulani Maphosa[2], Mitch Matoga[1,3]

1 Malawi HIV Implementation Scientist Training Program, Lilongwe, Malawi, 2 Elizabeth Glaser Pediatric AIDS Foundation, Lilongwe, Malawi, 3 UNC-Project, Lilongwe, Malawi, 4 University of Malawi, Chancellor College, Zomba, Malawi, 5 Institute for Participatory Engagement and Quality Improvement (IPEQI), Lilongwe, Malawi, 6 Queen Elizabeth Central Hospital, Ministry of Health, Blantyre, Malawi, 7 Kamuzu University of Health Sciences, Blantyre, Malawi, 8 University of Ghana School of Public Health, Accra, Ghana, 9 Amsterdam Institute for Global Health and Development (AIGHD), Amsterdam, The Netherlands

* rachelchamanga@gmail.com

**Data Availability Statement:** Our article contains quotes that were collected from health care

## Abstract

Following the COVID-19 pandemic, the Malawi Government released a policy that promoted the scale-up of six-monthly multi-month dispensing (6-MMD) of antiretroviral therapy (ART) to people living with HIV in order to decrease congestion at health facilities and transmission of COVID-19. We evaluated the barriers and facilitators to implementing the scale-up of 6-MMD. We conducted a cross-sectional study and collected quantitative and qualitative data from 13 January 2022 to 5 February 2022 at two high-volume primary health facilities in urban Blantyre, Malawi. A survey was self-administered to healthcare workers (HCWs) and a subset were purposively selected for key informant interviews. The interviews were guided by the consolidated framework for implementation research and questions focused on perceived barriers and facilitators to 6-MMD. We calculated proportions of reported barriers and facilitators based on the Likert scale. A thematic content analysis was done for qualitative data. Of the 77 HCWs who participated in the surveys, 56 (73%) were female and 22 (29%) were nurses. Major barriers to the implementation of 6-MMD were low drug supply and lack of understanding of the policy. Other reported barriers were s missing clinic appointments and viral load sample collection, if timelines for ART dispensation and viral load testing were misaligned. The major facilitators were orientation and review meetings, teamwork among staff, and the use of the electronic medical records system to track patients. Additionally, reduction in the number of patient visits, which reduced the workload of healthcare workers was cited as another motivator for implementing 6-MMD. Major facilitators to transition to 6-MMD included health care worker capacity building, teamwork and use of electronic medical records while major barriers included low drug supplies and lack of understanding of policy guidance. These findings may be helpful when developing strategies for increasing coverage and uptake of 6-MMD of ART.

workers during the key informant interviews. We believe that the key informants can be easily identified if the data were made public as their selection to take part in the study was based on their roles. Furthermore, at the time of ethical approval we documented that we would keep the study data confidential and not share with others who were not part of the study; we did not specify that data would be made available in a public repository. Therefore making the data public would not be in line with the approved ethical application and the information that was provided to participants about the study and how their data would be used. For those requesting data, we can certainly share the data after access has been granted by Malawi National Health Sciences Research Committee (NHSRC). Interested individuals can send an email to the Chairman of the Committee at mohdoccentre@gmail.com.

**Funding:** This work was supported by the National Institute of Health (Malawi D43 M-HIRST Pilot Grant to RC; Malawi D43 M-HIRST Pilot Grant to DM; Malawi D43 M-HIRST Pilot Grant to SM). Authors AB, MN, KT, TM and MM did not receive funding from the grant. The funders had no role in study design, data collection and analysis, decision to publish, or preparation of the manuscript. No additional external funding was received for this study. The authors have no financial relationships or competing interests that could be perceived as influencing the results or interpretation of this research. Its contents are solely the authors' responsibility and do not necessarily represent the official views of the funders.

**Competing interests:** The authors have declared that no competing interests exist.

## Introduction

Malawi has an HIV prevalence of 8.6%, with approximately 960,000 people living with HIV [1]. Approximately 97.9% of people living with HIV (PLHIV) who know their status are receiving antiretroviral therapy (ART) [1]. ART is crucial in the control of HIV as it leads to viral suppression, which in turn leads a decrease in HIV transmission rate and improved quality of life for PLHIV [1,2]. However, during the COVID-19 pandemic there were disruptions in ART provision; reasons for this included temporary closure of ART clinics and fear of contracting COVID-19 infection in HIV clinics [3,4]. Such challenges made it difficult for PLHIV to receive ART at their routinely scheduled visits [3].

Globally countries had to develop innovative ways to ensure that ART was accessible during unpredictable periods of the pandemic. One key innovation was the use of multi-month dispensing (MMD), which allowed for the provision of ART for longer periods, resulting in decreased visits of patients to health facilities [5,6]. Worldwide, the World Health Organization advocated for less frequent ART medication pick-ups at 3 to 6 monthly intervals for stable clients on ART in 2016 [7]. In Africa, Ethiopia was one of the countries to implement 6-MMD in Africa as part of their Appointment Spacing Model (ASM) in July 2017 where clients were reviewed twice a year and received a six-months ART supply [8]. The use of MMD in Africa however expanded in Sub-Saharan countries funded by the United States President's Emergency Plan for AIDS Relief (PEPFAR) as provision of MMD was minimal program requirement in order to promote patient-centre differentiated care from 2019 [9].

In Malawi, six-monthly multi-month dispensing (6-MMD) was initially implemented in 2019 as a form of differentiated service delivery (DSD) for stable patients [10]. Stable patients were defined as those who met the following eligibility criteria: had been on ART for at least 6 months; were on first-line ART, had no side effects or opportunistic infections, had a suppressed HIV viral load, were adherent to their clinic visit schedules, and were at least 24 years of age [10]. In April 2020 following the inception of the COVID-19 outbreak, the eligibility criteria for 6-MMD were revised to allow patients receiving dolutegravir-based ART regimens who weighed more than 20 kilograms and had been on ART for at least 3 months to receive 6-MMD (8). With the relaxed policy, patients did not have to have a suppressed viral load (<1,000 copies), also those with adherence problems were not contra-indicated to receive the 6-MMD (8). The relaxed policy however excluded pregnant and breastfeeding mothers and infants (who were to receive 3-MMD) (8). This revision of the policy was re-enforced in updated guideline in Jan 2021 and allowed for more people to receive 6-MMD of ART and led to decongestion of HIV clinics as a means of COVID-19 prevention [6,11,12].

While 6-MMD appeared to be a good intervention to prevent COVID-19 transmission, there were reported barriers reported with its implementation, such as drug stock-outs, inconsistency in following the eligibility criteria among providers [13,14] refusal to receive 6-MMD due to patient desire to have frequent consultations with the provider, and fear of carrying large amounts of pills home [15].

Despite the challenges reported implementing 6-MMD, several studies have reported multiple benefits from this model, such as reduced workload due to reduced clinic visits, improved retention into care, decrease transport costs, more free time to do other chores and commitments, and improved quality of care provided to patients [14,16–19].

However, the implementation of scaling-up the 6-MMD strategy has not been evaluated during the COVID-19 pandemic in Malawi. We evaluated the barriers and facilitators of implementing the 6-MMD strategy to better understand its implementation during the COVID-19 pandemic among PLHIV receiving care at HIV clinics in Malawi.

## Methods

### Study design

This was a cross-sectional study which triangulated findings from quantitative surveys with complementary qualitative key informant interviews.

### Study setting, population, and sampling

The study was conducted in two high-volume primary health care facilities in Limbe and Mpemba in Blantyre Malawi from 13 January 2022 to 5 February 2022. Both facilities have been pilot sites for 6-MMD for stable patients since May 2019, before the policy change in April 2020. Both sites scaled-up the rollout of the broadened 6-MMD strategy in April 2020, following the policy revision due to the COVID-19 pandemic. At Mpemba Health Centre about 78% of the patients had been transitioned to 6-MMD while at Limbe Health Centre about 80% of the patients had been transitioned to 6-MMD at the time of the study.

The study population included healthcare workers who worked at the HIV care clinics in the two health facilities. We included health care workers who provided written consent and were currently working in an HIV care clinic. Healthcare workers included nurses, clinicians, HIV Diagnostic Assistants (HDA), ART Clerks, Expert Clients (EC), HIV counsellors, and other HIV care workers. At the time of the study there were approximately 100 health care workers in the 2 health facilities. We administered surveys to healthcare workers and conducted interviews with key informants. For the healthcare worker surveys, we included all healthcare workers who were available to complete the survey questionnaire. For the key informants, a subset of healthcare workers was purposively selected based on their roles, and we included a representative from each cadre who consented and was available to have the key informant interviews.

### The framework of analysis

We used the Consolidated Framework for Implementation (CFIR) to explore the barriers and facilitators of multiple levels of care [20]. The CFIR classifies barriers/facilitators into five categories: characteristics of the intervention, inner setting, outer setting, individuals involved, and the implementation process. For our study, we considered the characteristics of the intervention to include whether it was complex and difficult to implement. We considered the characteristics of the individuals to include such things as knowledge of 6-MMD policy and attitude of healthcare workers regarding the policy. We considered the barriers and facilitators of the inner setting as those involving host institutional factors at the hospital level, such as teamwork or availability of resources in the hospital to implement the policy. The barriers and facilitators of the outer setting were those beyond the health facility and associated with the health system, such as drug stockouts.

### Data collection

**Quantitative.**   After written consent, healthcare workers completed a self-administered questionnaire in their preferred language (Chichewa or English). The questionnaire was self-administered using the Open Data Kit (ODK) which was loaded into an electronic tablet. Healthcare staff chose their preferred location within the health facility where they completed the questionnaire. The questionnaire had a combination of structured questions and a three-point Likert scale (agree, neutral, disagree), and focused on questions on perceived experiences of implementing the 6-MMD strategy, guided by the CFIR. Survey questionnaires were downloaded from the tablets into a comma separate value (CSV) file for analysis.

**Qualitative.** Purposively selected health workers who provided written consent to participate in key informant interviews had 30-60-minute audio-recorded interviews conducted in the local language of Chichewa, using open-ended questions guided by the CFIR that focused on perceived barriers and facilitators of implementing the 6-MMD strategy. We conducted 18 key informant interviews. The interviews were conducted in a private location in a health facility. The data collected from the audio recordings were appropriately labelled and filed electronically. The interviews were translated and transcribed into English. The Principal Investigator reviewed the transcripts as they became available and provided feedback to the interviewer during the period of data collection.

### Analysis

**Quantitative.** The CSV file with the healthcare worker surveys was imported into Statistics and Data (STATA) software. Demographic characteristics of the respondents were analyzed. Analysis of the scores using the Likert scale was performed, with results reported as proportions and summarized in tables.

**Qualitative.** We used a deductive approach to develop salient themes based on CFIR. In the initial phase, two coders independently coded 38% of the transcripts and agreed on standard themes and codes to ensure intercoder reliability. Double coding of the transcripts was performed using NVIVO 12, and any discrepancies were resolved to ensure consistency in coding among the research team. An intercoding Kappa Coefficient of 0.90 was used as the cut-off of the coding agreement. A thematic codebook was then developed in line with the themes explored in the interview guides as well as additional codes that emerged from key concepts in the participant responses based on the CFIR constructs. Thematic analysis was conducted to identify patterns across the data. We also conducted a coding matrix to determine the frequency of the themes. Finally, data were abstracted from the quotations of the respondents and included in the manuscript.

### Ethical considerations

Ethical approval was obtained from the National Health Sciences Committee (NHSRC) Institutional Review Board under protocol number 21/06/2666 to conduct this study. We also received approval from the Blantyre District Health Office to conduct the study in the two facilities. All the healthcare workers who enrolled in the study provided written informed consent. Additional information regarding the ethical, cultural, and scientific considerations specific to inclusivity in global research is included in the (S1 Text).

### Results

### Quantitative component

A total of 77 healthcare workers participated in the survey (Table 1). The majority (56, 73%) were female and nurses (22, 29%) by profession.

Table 2 shows the perceived effects of 6-MMD, as expressed by healthcare workers. Ninety-six percent (n = 74) of healthcare workers expressed that transitioning people to 6-MMD helped to reduce patient visits to the clinic and reduce crowding at the health facility; 88% (n = 68) noted that the 6-MMD strategy helped reduce the workload at the health facility. However, 23% (n = 18) of the healthcare workers reported that they thought the 6-MMD strategy led to loss to follow-up of people on ART, with 69% (n = 53) of the healthcare workers noting that transitioning people to 6-MMD requires interventions such as telephonic visits to promote adherence and/or avoid loss to follow-up. Furthermore, 29% (n = 22) of the

**Table 1. Characteristics of participant in the healthcare worker surveys.**

| Healthcare worker survey, N = 77 participants | | |
|---|---|---|
| **Characteristics** | **Description** | **n (%)** |
| Sex | Male | 21(27.3) |
| | Female | 56(72.7) |
| Mean age (SD) | 34 (+ 7) | |
| Professional Cadre | Clinician | 7 (9.1) |
| | Nurse | 22 (28.6) |
| | HIV Diagnostic Assistant | 10 (12.9) |
| | Expert Client | 11 (14.3) |
| | Psychosocial counsellor | 3 (3.9) |
| | Data Clerk | 5(6.5) |
| | Health Surveillance Assistant | 13(16.9) |
| | Youth Champion | 4(5.2) |
| | Pharmacy assistant | 1(1.3) |
| | Adherence Support Officer (ASO) | 1(1.3) |
| Mean years in role (SD) | 6 (±4) | |

healthcare workers thought that the new policy allowed for unstable patients to transition to 6-MMD, including those with high HIV viral load who would normally require close follow-up, while 43% (n = 33) disagreed with this suggestion. Nevertheless, 92% (n = 71) of the healthcare workers reported that if the COVID-19 pandemic stabilized, they would advocate for

**Table 2. Experiences of conducting six-monthly multi-month refills as scored by healthcare workers.**

| | Disagree | Neutral | Agree |
|---|---|---|---|
| Transitioning people to 6-monthly refill has helped to reduce crowding at the health facility | 1(1.3) | 2(2.6) | 74 (96.1) |
| Transitioning people to 6-monthly refill has helped to reduce patient clinic visits at the health facility | 2(2.6) | 1(1.3) | 74 (96.1) |
| Transitioning people to 6-monthly refill has helped to reduce workload at the health facility | 3(3.9) | 6(7.8) | 68 (88.3) |
| Transitioning people to 6-monthly refill has helped health facilities to be able to still deliver essential health services during the COVID-19 pandemic | 1(1.3) | 6(7.8) | 70 (90.9) |
| Transitioning people to 6-monthly refill has led to unstable patients e.g people with high HIV RNA viral load being provided 6 months' supply which would not have been the case before | 33(42.8) | 22 (28.6) | 22 (28.6) |
| Transitioning people to 6-monthly refill has led to loss to follow up of people on ART | 40(51.9) | 19 (24.7) | 18 (23.4) |
| Transitioning people to 6-monthly refill requires use of other interventions such as telephonic visits to promote adherence or avoid loss to follow up | 17(22.1) | 7(9.1) | 53 (68.8) |
| All my patients are comfortable to carry the 6-monthly supply and do not complain of having multiple bottles to carry. | 4(5.2) | 6(7.8) | 67 (87.0) |
| Although people are transitioned to 6-monthly ART refill, some people still show up for unscheduled visits because they are sick | 13(16.9) | 11 (14.3) | 53 (68.8) |
| The restrictions released by government e.g. social distancing, etc had an effect on people's general ART attendance and so we have a backlog of participants on 3-monthly refills who have not showed up to be transitioned | 36(46.7) | 16 (20.8) | 25 (32.5) |
| Even if COVID-19 pandemic stabilizes, I would advocate that we continue the current practice of 6-monthly ART refills because it is beneficial to the patients and health facility staff | 2(2.6) | 4(5.2) | 71 (92.2) |

continued implementation of the 6-MMD strategy, as it is beneficial to patients and health facility staff.

## Qualitative component

A total of 18 key informant interviews were conducted (Table 3). Most of the participants (67%) were female. Clinicians were the most common cadre interviewed (7/18). The mean age for the respondents was slightly higher than those that took part in the survey (38 years vs 34 years) (Table 3).

## Barriers and facilitators to 6-MMD

Table 4 summarizes the reported barriers and facilitators identified in this study. Under the three CFIR domains (characteristics of the individuals, outer setting, and inner setting), healthcare workers reported several barriers, including: inadequate knowledge of 6-MMD policy by facility staff; low stocks of either cotrimoxazole or antiretroviral therapy; lack of viral load results; lack of viral load testing supplies; and fear of COVID-19 vaccination being required to attend the clinic. Other barriers that were also perceived as negative effects of the policy included: delayed or missed visits by patients and missing collection of viral loads if the timelines for ART pick-up and viral load testing were not aligned.

The facilitators that were reported included: acceptance of the policy by healthcare workers; teamwork; use of electronic medical records system; conducting orientation meetings; prior experience in conducting the 6-MMD as a differentiated service; and packaging of the drugs into a bottle with a 3-month supply. Additional facilitators, which were also viewed as benefits, included reduced patient visits and reduced crowding at the facility; reduced workload for healthcare workers; reduced transport costs for patients as they come fewer times; patients having free time to perform other tasks such as other income generating activities; improved quality of care to patients as healthcare workers are able to spend enough time with each patient; improved retention into care of patients; and protection from COVID-19.

**Table 3. Demographic characteristics of health care workers who provided key informant interviews.**

| Key informant interviews, N = 18 participants | | |
|---|---|---|
| | **Characteristics** | **n(%)** |
| Gender | Male | 6 (33) |
| | Female | 12 (67) |
| Professional Cadre | Clinician | 7 |
| | Nurse | 3 |
| | HIV Diagnostic Assistant | 1 |
| | Expert Client | 2 |
| | Data Clerk | 1 |
| | Youth Champion | 1 |
| | Adherence Support Officer | 3 |
| Mean age (SD) | 38 (6) | |
| Mean years worked | < 1 year | 1 |
| | 1–5 years | 7 |
| | 6–10 years | 6 |
| | >10 years | 4 |

**Table 4. Summary of barriers and facilitators identified in this study.**

| CFIR domain | Barriers or Facilitator | CFR construct or theme | Description |
|---|---|---|---|
| Outer setting | Barrier | Low supply of ART drugs and cotrimoxazole | Low supply of ART drugs hindered the transition to 6-MMD |
| | Barrier | COVID-19 vaccination | Fear of COVID-19 vaccination led to less people willing to come to health facilities |
| | Barrier | Lack of viral load supplies | Lack of viral load supplies prompted the healthcare workers to schedule clients sooner |
| | Facilitator | Drug packaging | Packaging the drugs for 3 months in 1 bottle led to less bottles for patients to carry |
| Inner setting | Barrier | Missing viral load results | Some healthcare workers utilized viral load results to decide on transitioning the patient and were unable to transition patients if results were missing |
| | Barrier | Misaligning viral load collection | Collection of viral load sample was missed/misaligned if healthcare workers provided 6-MMD instead of providing a supply to match the viral load collection date |
| | Facilitator | Teamwork | Working as a team ensure that healthcare workers were able to check on each other and ensure that correct processes were being followed |
| | Facilitator | Use of Electronic Medical Records | Electronic Medical Records provided a platform to track those transitioned to 6-MMD or not |
| | Facilitator | Orientation/ Review meetings | Orientation meetings provided initial knowledge of the policy. Review meetings helped to track the progress in implementation of the policy |
| | Facilitator | Prior experience in conducting of MMD | Prior experience in conducting 6-MMD at the facility helped to scale-up its implementation |
| | Facilitator | Reduced workload | Reduced workload for healthcare staff due to reduced number of patients seen per day allowed for healthcare staff to do other tasks |
| | Facilitator | Improved patient care | The reduced workload led to improved patient care as healthcare workers had more time to attend to individual patients |
| | Facilitator | Improved retention into care | The patients were more likely to come on the specified date if visits were more spaced out and this improved retention of the patients |
| Characteristics of individuals | Barrier | Lack of knowledge | Lack of knowledge about the policy led to transition of ineligible people |
| | Barrier | Delayed or missed patient visits | Some patients delayed or missed their scheduled date for ART refill and clinical review as they still had supply of drugs through 6-MMD |
| | Facilitator | Acceptance of the policy by health workers | Acceptance of policy provided motivation to implement the policy |
| | Facilitator | Reduced number of patient visits | Reduced number of patient visits allowed patients to save transport costs |

## Barriers

### A. Characteristics of individuals

**Lack of understanding of the policy.** Despite orientation meetings at both facilities, mis-understandings about the eligibility criteria for the 6-MMD were apparent among healthcare workers, which resulted in persons who did not meet the eligibility criteria being transitioned to 6-MMD.

*The other challenge has been on the provider side, perhaps not understanding the criteria and ending up putting a person who was not supposed to be on six months, Let's say the person with high viral load, you are supposed to do three months intensive adherence counselling, so, it happens that the person did not see the viral load results, you have just transition him/her to six months. So, it means we have missed that person, we will not follow him/her up (Mpemba_Nurse_6)*

## B. Outer settings

**Fear of getting COVID-19 vaccine.**    Fear of getting a COVID-19 vaccination may have prevented several patients from going to the health facility to get their ART refills, which meant there were fewer people who made themselves available to be transitioned to 6-MMD.

*Another thing is that at some point, people were so much afraid to come to the clinic because of COVID. That interfered . . . because they were refusing. "you will first get COVID jab before receiving drugs" that was a very famous rumor that time,. . . yeah,so, many were afraid to get the drugs, yeah that's it, But now, people discovered that those were mere lies, they now know the truth, they have started coming, the big challenge was during that time.*(Limbe_ Clinician7).

**Low drug supply.**    Problems with a decreased ART drug supply at certain points during the implementation period of the policy significantly impacted the implementation of the 6-MMD strategy. This was the most commonly reported barrier by 9 of 10 ART prescribers (clinicians and nurses). In both facilities, ART shortages forced providers to devise innovative ways of sustaining patient drug supply, including rationing the supplies to be able to provide ART to all patients. A low stock of ART drugs was reported from October to December 2020 in the two facilities.

*The supply we got was very few, compared to the ratio of the clients we have. There was a memo, the issue was to do with, shipment. Since Covid, this wave started from countries which are manufacturing the drugs such as India, and others, they were hardly hit with Covid.. So, it seems that, their production that side was very low. . .. I should say, the supply got reduced. . .. So, when the supply was coming, if we get the ratio of clients we have in a quarter, aah, versus the bottles we have, it was discovered that, if we were to give the 6 months, eventually, the drugs, it could have reached a point where we could experience a shortage. . .. so the ministry said, let's aah, should reduce the 6 months prescription. They didn't say stop, but reduce the 6 months prescription. So that, at least everyone, there should not be anyone completely without the drugs* (Limbe_Clinician_5).

In addition to the low supply of ART drugs, a low supply of additional drugs was also reported. Cotrimoxazole (also known as Bactrim) preventive therapy (CPT) is used for prevention of Pneumocystis pneumonia and bacterial infections in PLHIV, and was the most commonly used additional drug in HIV patients. The shortage of cotrimoxazole also led to patients needing to frequently visit the clinic even if ART was available for 6-MMD supply.

*Recently, or even now we have the challenge of Bactrim. So, there is no bactrim and what happens is that if it comes in short supply it means they will be getting 13A for six months but the bactrim will be few because we want to carter for many people so we will be telling them that once they finish this one you will come back. (Mpemba_Nurse_6)*

**Lack of viral load supplies.**    Lack of adequate materials to collect and conduct viral load tests, such as lack of dried blood spot bundles supplied by the government, was highlighted as another challenge in preventing staff from transitioning patients to 6-MMD, as they could not conduct the required patient viral load test and hence needed the patient to come sooner when the supply was restored so that their viral load could then be tested.

*Also, another challenge is that we had no bundles for collecting viral load so we failed to give them six months because most of them were eligible for viral load collection. So, we feared that if we give them 6 months, when the bundles arrive it would happen that the person has gone. So this limited us to give them three months' supply. (Limbe Nurse_3)*

## C. Inner settings

**Missing viral load results.** Despite there being a relaxed criterion, where the policy said availability of viral load results was not required to transition to 6-MMD, healthcare staff reported that were not able to transition patients to 6-MMD if the viral load result was not available. This made it difficult for the ART provider to decide whether to transition patients when the patient had come for ART dispensing.

*Sometimes it happens that instead of taking good care of the results that have come, you find that there are no results in the card or file. And you start following-up for results which delays the patient. At times you do not find the result and you have to take another sample and its painful for you to be getting their sample again and again. I think what is needed is that if we get a sample we should have a good follow up so that we should have the results right away when the client comes so that we should be able to decide whether to give six months or not. (Mpemba_Clinician_9)*

**Delayed or missed visits by patients.** Healthcare workers noted that some patients did not come to clinic on their scheduled appointment date because the patient still had ART drugs due to an additional supply that is contained in the bottle for 6-MMD, known as a "buffer supply." This led to patients being flagged as defaulters from care, which necessitated them to be traced by health facilities.

*When we give them 6 months, in each bottle.. there were 4 tablets which were remaining as their buffers, for the next appointment date. . . so, when we count for 6 months, if we count 4 times 6, it's almost 24, and if they happen to forget within, for 1 or 2 days, it means they will have 30 tablets which is almost a month. They won't come. . .. because they would have the drugs. But they would be considered as defaulters. So, when they come here, and you ask them why they have been defaulting, they would say that, "I have been taking the drugs" . . . Maybe if the bottles contained exact tablets (30), there would not be such cases. (Limbe_Clinician_7)*

**Missing viral load collection at milestones.** Healthcare workers noted that implementation of 6-MMD led to healthcare workers missing collection of viral load samples at the required time points for collection, especially if they did not align their drug refill schedule with the time when the viral load was supposed to be collected.

*The other challenge is that, it might be that the client we're seeing is supposed to have his/her viral load taken the next month you are seeing him/her, so it happens that you have overlooked that fact, and it happens that you have given six months, so, he/she will come almost four months after the date you (were supposed to) take the viral load.so, that one could also be a challenge, if you are not careful provider can miss out the most important milestones (Mpemba__Nurse_6)*

## Facilitators

### A. Characteristics of Individuals

**Acceptance by the patients and healthcare workers.** The success of the 6-MMD implementation was partly due to the receptive attitude of patients and providers towards the policy. In both Limbe and Mpemba facilities, it was reported that the policy was well accepted by patients and healthcare workers because of its perceived advantages at the time of introduction.

*the first main factor is that people accepted it well. So, when people receive something, there is no resistance. Both the provider and the patients, the clients, it was well acceptable. Everybody welcomed it and they were happy* (Limbe_Nurse_3).

**Reduced transport costs for patients and patients having time to do other activities.** Healthcare workers reported that they thought provision of 6-MMD dispensing motivated patients as patients had fewer visits and thus it reduced their transport costs as well as provided the patients with time to do other activities, which contributed to improved retention of the patients.

*They were very happy, they were very, very happy. Most had complaints with frequent visits that they spend on transport. . . now issues like those have been sorted out. They are coming to the hospital like twice a year. Others are shy. They don't like coming to the hospital often times to get medicine. So, when they are given the six months they rest. So, it also shows that the six months has improved on the retention rate. (Limbe_Nurse_3)*

### B. Inner setting characteristics

**Teamwork.** Teamwork was noted as a contributing factor to effective policy implementation. Working in teams was beneficial as it enabled staff to correct each other's mistakes and remind each other of the policy.

*"That of working together as a team at this facility. Because sometimes, a human being is a human being, I can miss, a mere person here, but when that one meets other colleagues, the person can be discovered that this one, so you can make alterations, but we also remind one another, that guys now we are using such, such policy."* (Limbe_Clinician_7).

**Use of the electronic medical records system ("J2 system").** Transitioning clients to 6-MMD required looking at patient information to identify and track the patients that were eligible to be transitioned. The electronic medical records "J2" which is a database of all the patients at the facility, was helpful in tracking clients who required to be transitioned as it showed the patient medical history and confirmed eligibility.

*the "J2 system" since whenever you punched in the 'ID' number for a particular person, it shows everything, like "this one has been for such months, on ART" so, even the client would*

*not lie to you, when you check in that, that's it. So, I see that helped us a lot, that it should not be difficult (Limbe_Clinician_7)*

**Orientation meetings.** ART providers explained that they conducted orientation meetings in their facilities to acquaint all ART staff members with the 6-MMD policy at the time of its advent. This was particularly important, as formal training was not conducted.

*I think that was a main factor which drove us more, besides, information came, they trained us on the 6 months dispensing, about transitioning, so, people were given information. . .at, the ART staff, training us that we will be transitioning, from there, to, to 6 months, for these reasons. . .. it wasn't a training as such, but it was just an orientation. (Mpemba_Clinician_9).*

In addition to orientation meetings, regular review meetings were reported by both facilities. Review meetings aimed at tracking the progress of implementing the 6-MMD policy. More importantly, the meetings were said to provide a platform for providers to present and discuss the challenges encountered in the implementation process of the policy in the facilities. Providers also reported that review meetings served as avenues for staff members to share and exchange knowledge and experiences regarding the 6-MMD policy.

*We also have review meetings, monthly review meetings. So, with that, we point out some of the gaps that we have. So, when we do that, we make changes, like, maybe when we have identified that, we have a certain gap, we discuss the solution, that, "what should we do with this gap, aah, we should do like this and that, and that" (Mpemba_Nurse_7)*

**Prior experience in conducting 6-MMD as a DSD.** Prior experience in conducting the 6-MMD in stable patients was noted as a facilitator in being able to conduct the 6-MMD even in unstable and stable patients after the COVID-19 pandemic.

*I would say that it helped us, since ..we already knew the things which we consider, to be this and that, eeh, when COVID came, it was not hard, since we already know that if one has these, he/she should be transitioned to 6 months, if one doesn't have these, then he/she is not supposed to be transitioned to 6 months. (**Limbe_Clinician_7**)*

**Reduction in patient visits and protection from COVID-19.** The reduction in the number of patient visits, which led to reduced workload and reduced crowding at the facility, was a major benefit and motivating factor for healthcare workers in implementing the 6-MMD. Healthcare workers noted that this helped them to provide improved patient care.

*"It reduces the number of people whom we are seeing per day, by doing that we are having enough time because . . .let's say you have 20 clients or 10 clients, you have ample time to listen to each and everybody compared to the day you have 60 clients. . ..Because sometimes you might be treating the queue other than the individual problems that the person has brought, so, when you look at, in terms of work load it has reduced the work load which has also improved the quality of care that we are providing to the clients, so, I like it. (Limbe_Clinician7).*

With patients coming in fewer numbers, healthcare workers felt the risk of contracting COVID-19 was reduced.

*"It reduced workload since people come in a few numbers and also because people come in a few numbers even the risk of us contacting COVID is minimized but back then with a lot of people it was easy for someone to get Covid-19.(Mpemba_Nurse_6)*

## C. Outer setting

**Drug packaging.**   The change in packaging drugs from each bottle having a monthly supply to then having a 3-monthly supply was cited as a facilitating factor for moving to 6-monthly dispensing, as it led to patients carrying fewer bottles of drugs.

*I can say that after we started giving 3 months, they brought us 13 A one bottle containing 90 tablets. So the person was no longer taking so many bottles. . . ... At first before those medicines came it would mean that they (clients) would take 6 bottles but now they only take 2 bottles for 6 months (Limbe_Nurse_3).*

## Discussion

In this study of the implementation of the 6-MMD strategy, which was guided by a COVID-19-related policy applicable to both stable and non-stable patients, major barriers to the implementation of 6-MMD were low drug supply and lack of understanding of the policy. Other reported barriers were that the provision of 6-MMD led to delayed or missed clinic visits; and missing viral load sample collection if the healthcare worker misaligned their viral load sample and clinic visit timelines. The major facilitators associated with the provision of the 6-MMD strategy were orientation and review meetings, teamwork among staff and use of the electronic medical records system to track patients. The reduction in the number of patient visits, which reduced the workload of health care workers and led to improved care provided to patients, were cited as other motivators for implementing the 6-MMD.

Most health care workers reported that low drug supply was barrier to implementing 6-MMD.A study conducted in South Africa showed that health care workers raised concerns on whether the supply chain was reliable to manage extended multi-month dispensing [21]. Prior studies conducted in Malawi have shown that drug stockouts influence the roll-out of longer periods of drug dispensing to multi-month dispensing [14,21]. The situation of low drug supply may have been more frequent in the COVID-19 era, as challenges such as shipping of drugs were reported by manufacturers [22,23]. Nevertheless, in order for sustained provision of MMD, adequate supply of ART is an indispensable requirement.

Inadequate knowledge of healthcare workers was noted as another barrier that affected implementation of the policy. Interestingly despite that the policy allowed 6-MMD for all clients irrespective of viral load test result; lack of viral test result continued to be reported as a barrier even after the new policy was released. Furthermore, in the health care survey questionnaires, most of the participants did not think that the new policy led to transitioning of unstable patients. This discrepancy could have been because the new COVID-19 guideline may not have been followed as required due to lack of understanding or health care workers still felt the need to check the patient's viral load before transitioning them.

Prior studies conducted in Malawi have shown that healthcare workers commonly confused the eligibility criteria for those requiring three-monthly MMD, community ART, or fast-

track refills due to limited healthcare workers' knowledge of the requirements [13]. Studies conducted in the African setting have also shown low uptake of DSD models when the provider's literacy of the model is low [24]. In this study, facility-orientation meetings followed by continuous review meetings seemed to improve knowledge of health care workers. As such, there is need for continuous reviews to address the knowledge gaps that occur when new policies are introduced.

Misconceptions about COVID-19 vaccinations led to a decrease in the number of people coming to health facilities. Prior research has shown that there was a decrease in the uptake of HIV care services during the COVID-19 pandemic in Malawi due to the lockdown of the country secondary to the pandemic and a resultant slowdown in the provision of HIV services; however, the attribution of low ART clinic attendance due to misconceptions that COVID-19 vaccine was required to be administered when attending a clinic visit is a new finding [23].

Malawi predominantly uses a paper-based system in its health facilities. Electronic medical records, which were newly introduced in Malawi, were noted as enablers in implementing 6-MMD. Research in Malawi has shown that electronic medical records assist in prompt decision-making and provision of care to people on ART, and other studies have utilized electronic medical records to evaluate the uptake of MMD [13,25]. The electronic medical record being noted as an enabler in the provision of care is encouraging and advocates the deployment of such systems in Malawi.

Prior experience in conducting 6-MMD as a DSD model was shown to be helpful in ensuring scaling up of the 6-MMD to other populations during the COVID-19 pandemic. This is encouraging and demonstrates how a DSD model can be scaled-up if necessary to address critical health needs when unprecedented events such as pandemics occur [19].

Interestingly, drug packaging was shown to be a facilitator for 6-MMD, as it led to fewer bottles required to be carried by a patient. This preserves the anonymity of a patient's HIV status as patients who are on 6-MMD worry that their status will be disclosed because they carry a lot of bottles which make noise once in their bag, and may prompt queries from colleagues [21]. Other studies have shown that providers are concerned about patients having challenges in collecting a large number of bottles and storing them [16,21]. However, the same studies showed that clients did not have any concerns about challenges with storing drugs contrary to the perceptions of health care workers. Our study suggests that packaging of drugs for several months in one bottle may be beneficial when considering implementation of MMD, as it decreases the number of bottles provided to clients.

The other facilitators of 6-MMD reported in this study, such as reduced workload and patient visits, are similar to those reported elsewhere [14,16]. Negative effects, such as the delay in patients reporting to the health facility for clinic visits leading to their misclassification as lost to follow-up, have also been reported [14]. However, contrary to this health care workers noted that they felt that there was an improvement in retention into care at the facility. Studies done in Mozambique and Haiti have also shown than provision of ART for longer intervals were associated with improved retention into care and not loss to follow up [26,27]. Our findings showed that the implementation of this intervention requires healthcare workers to provide reminders to patients. The decrease in buffer drugs is also a consideration that needs to be taken when rolling out the 6-MMD, which can be controlled if standard drug packaging is used for ART supply of longer months.

We believe that the strength of this study is the use of quantitative and qualitative methods to better explain our findings by triangulating results. Furthermore, the use of the CFIR model allowed us to systematically and holistically assess barriers and facilitators of 6-MMD. The limitations of the study were that the sample was from urban health facilities; hence, the challenges discussed may not be applicable to semi-urban or rural settings. Additionally, inclusion of

patient experience could have provided perspectives which could have been helpful to the study. Lastly, inclusion of facilities already conducting the 6-MMD may have caused some bias. Nevertheless, our results can guide decision-makers in developing protocols for implementing 6-MMD in multiple settings.

## Conclusion

This study showed that low supply of drugs and lack of understanding of the policy were the major challenges that inhibited transitioning people to 6-MMD in both stable and unstable patients amidst the COVID-19 pandemic. Facilitators for the 6-MMD included orientation and review meetings, teamwork among staff, and the use of the electronic medical records system to track patients. The barriers and facilitators elicited in this study should be considered when developing strategies for increasing the coverage and uptake of 6-monthly dispensation of ART.

## Supporting information

**S1 Text. Inclusivity in global research questionnaire.**
(PDF)

## Acknowledgments

We would like to acknowledge the health care workers from Mpemba and Limbe Health Centres who took part in the study.

## Author Contributions

**Conceptualization:** Rachel Chamanga, Mulinda Nyirenda, Kwasi Torpey, Mitch Matoga.

**Data curation:** Rachel Chamanga, Denview Magalasi, Stella Mahuva.

**Formal analysis:** Rachel Chamanga, Agatha Bula, Denview Magalasi, Stella Mahuva.

**Methodology:** Rachel Chamanga, Denview Magalasi, Thulani Maphosa, Mitch Matoga.

**Supervision:** Agatha Bula, Thulani Maphosa, Mitch Matoga.

**Writing – original draft:** Rachel Chamanga, Kwasi Torpey, Thulani Maphosa, Mitch Matoga.

**Writing – review & editing:** Rachel Chamanga, Kwasi Torpey, Thulani Maphosa, Mitch Matoga.

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
