## [Decision Letter · Decision Letter 0]

2 Aug 2024

PGPH-D-24-01355

Barriers and facilitators to implementing six monthly multi-month dispensing of antiretroviral therapy in HIV clinics during the COVID-19 Era in Malawi

Dear Dr. Chamanga,

Thank you for submitting your manuscript to PLOS Global Public Health. After careful consideration, we feel that it has merit but does not fully meet PLOS Global Public Health’s publication criteria as it currently stands. Therefore, we invite you to submit a revised version of the manuscript that addresses the points raised during the review process.

We look forward to receiving your revised manuscript.

Kind regards,

Tsitsi B. Masvawure, Ph.D.

Academic Editor

Journal Requirements:

2. We have amended your Competing Interest statement to comply with journal style. We kindly ask that you double check the statement and let us know if anything is incorrect. 

3. Your current Financial Disclosure states, “The study was supported by the Malawi D43 M-HIRST Pilot Grant from the Malawi HIV Implementation Research Scientist and Training Program”. However, your funding information on the submission form indicates that you received funding from “Funder name is Foundation for the National Institutes of Health with grant number D43 and grant recipient Dr Rachel Chamanga”. Please indicate by return email the full and correct funding information for your study and confirm the order in which funding contributions should appear. Please be sure to indicate whether the funders played any role in the study design, data collection and analysis, decision to publish, or preparation of the manuscript.

4. In the online submission form, you indicated that "The research data used in this study is readily available to share with the journal upon request from the lead author. The data is stored in the lead author`s Google Drive."

a. In a public repository, 

b. Within the manuscript itself, or 

c. Uploaded as supplementary information.

Additional Editor Comments (if provided):

Thank you for your submission. I invite you to respond to the comments from all three reviewers. Most of the suggested changes are relatively minor.

Reviewers' comments:

Reviewer's Responses to Questions

**Comments to the Author**

1. Does this manuscript meet PLOS Global Public Health’s publication criteria? Is the manuscript technically sound, and do the data support the conclusions? The manuscript must describe methodologically and ethically rigorous research with conclusions that are appropriately drawn based on the data presented.

Reviewer #1: Yes

Reviewer #2: Yes

Reviewer #3: Partly

2. Has the statistical analysis been performed appropriately and rigorously?

Reviewer #1: Yes

Reviewer #2: Yes

Reviewer #3: No

3. Have the authors made all data underlying the findings in their manuscript fully available (please refer to the Data Availability Statement at the start of the manuscript PDF file)?

Reviewer #1: Yes

Reviewer #2: Yes

Reviewer #3: No

4. Is the manuscript presented in an intelligible fashion and written in standard English?

Reviewer #1: Yes

Reviewer #2: Yes

Reviewer #3: Yes

5. Review Comments to the Author

Reviewer #1: Summary:

A cross-sectional study done in two high volume urban health facilities in Malawi using qualitative and quantitative data to review barriers and facilitators of six monthly multi month dispensing of antiretroviral drugs during COVID 19 era. Major barriers to implementation were low drug supplies and lack of understanding of policy guidance. The author recorganizes that implementation of 6-MMD may have led to cases of delayed or missed clinic appointments and missed opportunities for viral load sample collection if clinic appointments were not aligned. Major facilitators of 6-MMD were health care workers orientation and review meetings, teamwork among staff, and use of the electronic medical records system to track patients. Additional motivators to implementing 6-MMD included reduction in the number of patient visit and reduced workload at health facilities. Study findings may be relevant in the development and implementation policies on multi month dispensing of ART among HIV positive clients.

Overall comments

- Review document and remove any double spaces.

- Review In text referencing for consistency. E.g …..visits(3). …… facilities. (5, 6) …….. patients (7). care, (12, 13) (14, 15),

- Minor spelling errors

1. Title: Barriers and facilitators to implementing six monthly multi-month dispensing of

antiretroviral therapy in HIV clinics during the COVID-19 Era in Malawi

Comment: Title seems to suggest that findings are generalizable to whole population while this was a descriptive study limited to two high volume health facilities in an urban setting. Please review

Consider: Barriers and facilitators to implementing six monthly multi-month dispensing of antiretroviral therapy in two urban /selected HIV clinics during the COVID-19 Era in Malawi.

2. Abstract:

Summary is well presented and includes key findings from the study.

a. Conclusion section needs review: Looks like a repetition of the findings. Consider revising and include only key findings from the study. Avoid “commanding” language e.g … should… Suggestion: Major facilitators to transition to 6-MMD included health care worker capacity building, teamwork and use of electronic medical records while major barriers included low drug supplies and lack of understanding of policy guidance. These findings may be helpful when developing strategies for increasing coverage and uptake of 6-MMD of ART”

3. Introduction

a. Include information/background on the two pilot facilities: Number of PLHIV on treatment, Changes in number of clients on 6MMD..…… This may help put the narratives into context

b. ……Decreased rate of transmission and longer quality of life…..

Comment: Consider rewording to …”Decrease in HIV transmission rate and improved quality of life”….

c. Suppressed viral load

Comment: Please include reference value (? <50 copies/ml ?200 copies/ml ?<1000 copies/ml)

d. In April 2020 following the inception of the COVID-19 outbreak, the eligibility criteria for 6-MMD were revised to allow patients receiving dolutegravir-based ART regimens who weighed more than 20 kilograms and had been on ART for at least 3 months to receive 6-MMD (8)

Comment: Did this include both those newly initiated on DTG and those already on ART but transitioned to DTG based on availability of new molecules? Please clarify/add

e. With the relaxed policy, patients did not have to have a suppressed viral load, also those with adherence problems were not contra-indicated to receive the 6-MMD (8)

Comment Consider including a statement that although high VL and patient adherence challenges were not a contraindication to 6-MMD, adequate support systems were put in place to ensure patients achieved viral suppression.

4. Methods: I request that the section be reviewed by a subject matter expert preferably a behavioral scientist with support from a statistician.

5. Results: I request review by subject matter expert preferably a behavioral scientist with support from a statistician

a. Question: Some countries recommend 6 monthly clinic consultation, but 3 monthly ART pick up by clients. It looks like policy guidance in this case was slightly different. Please confirm.

b. Consider adding absolute numbers to the narrative given that the absolute numbers (Ref Table 2) are small, to put the percentages into context for the reader. E.g 96% (n=xx).

c. Minor typing errors: Loss of follow up:

Comment: Did you men lost to follow up?

d. Table 2: Transitioning people to 6-monthly refill has led to unstable patients e.g people with high HIV RNA viral load being provided 6 months’ supply which would not have been the case before

Consider adding in discussion section that despite policy guidance that allowed 6-MMD for all clients irrespective of viral load test result among others, one would have expected most respondents to agree with the statement which was not the case. Either previous guideline may not have been followed as required and a few unstable patients were getting 6-MM or respondents did not understand the question.

6. Discussion: Additional inputs from a behavioral scientist would be helpful

a. …….. Dispensing of drugs to clients who were not eligible for 6MMD such as those with high VL……

Comment: In the introduction section, it is stated that patients did not need to have a suppressed viral load to receive 6-MMD. The two seem to contradict each other. Please review.

b. Most references are based on studies done in Malawi. Review of study findings from other countries may enrich the discussion.

c. For consideration in discussion: Increase in number of clients not keeping clinic appointments may either be attributed to clients having more drugs or fear of coming to health facilities due to COVID 19. It is not clear from the discussion where there was an apparent increase in number lost to follow up. It could be true that there was an increase in number LTFU or what is a reporting issue. Additional literature may be helpful.

a. Drug packaging. …… Other studies have shown that providers are concerned about patients having challenges in collecting many bottles and storing them(12, 16).

Comment: Consider including client perspective in the discussion. Looks like in the two quoted references, responses from beneficiaries was slightly different ” …. They also expressed concerned about clients' challenges with ART storage at home, but clients reported no storage problems Ref 12. …….. Contrary to key informant concerns about patients' responsibility to manage larger quantities of ART, patients receiving 6-month refills were highly motivated and did not face challenges transporting, storing or adhering to treatment (ref 16).

d. Our study suggests that packaging of drugs for several months in one bottle should considered when implementing MMD

Comment: Consider rewording to … Our study suggests that packaging of drugs for several months in one bottle may be beneficial when considering implementation of MMD (Let us avoid commanding language)

Reviewer #2: I congratulate the authors for this perspective on how COVID-19 was intersecting with HIV care and retention, however, here are some minor revisions:

Abstract

1. "the six-monthly multi-month dispensing (6-MMD)" - The term "six-monthly multi-month" seems redundant. "Six-monthly dispensing (6-MMD)" should be sufficient.

2. "from 13 January 2022 to 5 February 2022" - Consistency in date format is important. It would be clearer to use either "January 13, 2022" or "13 January 2022" throughout the document.

3. "conduction of orientation meetings" - The correct term should be "conducting orientation meetings."

Methodology and discussion section

1. "Orientation and review meetings, teamwork among staff, and the use of the electronic medical records system" instead of "of orientation and review meetings, teamwork among staff and use of the electronic medical records system"- Corrected the phrasing for clarity and grammatical correctness.

2. "It was cited as another motivator" instead of "were cited as other motivators" - Corrected subject-verb agreement.

3. "Conducted in Malawi have shown"instead of "conducted in Malawi have showed" - Corrected verb tense.

4. "For those requiring three-monthly MMD, community ART, or fast-track refills due to limited healthcare workers’ knowledge of the requirements" instead of "those requiring three-monthly MMD, community ART, or fast-track refills because limited healthcare workers’ knowledge of the requirements" - Corrected phrasing.

5."Vaccine was required to be administered when attending a clinic visit" instead of "vaccine was required to be administered when attending a clinic visit is a new finding" - Corrected tense for clarity.

6."For other patients outside of the HIV arena"instead of "other patients outside of the HIV arena"- Corrected preposition usage.

7."Classified as lost to follow-up"instead of "being classified as lost to follow-up"- Corrected phrasing for conciseness.

8."Nevertheless, our results can guide decision-makers" instead of "Nevertheless, our results can guide decision makers"- Corrected punctuation.

Reviewer #3: The paper written by Chamanga et. al evaluated barriers and facilitators for providing six months duration antiretroviral therapy (ART) to people living with HIV (PLHIV) in Malawi at the peak of the COVID-19 pandemic. They also looked at the health care worker perspective on the effects of six monthly refills on the health system and clients.

The authors employed a mixed methods approach through a survey and key informant interviews to ten types of health care cadres working closely with PLHIV on ART at two health facilities in early 2022.

The paper is well written and easy to comprehend. While the survey also looked at the effects of 6-MMD (Table 2), these findings are not reflected in the title or the conclusion of the paper.

Specific comments for different sections are as follows:

Introduction

1. Reference 8, maybe used in conjunction with Edition 5, 26th January 2021 where routine viral load testing was suspended except for the 6 months point and targeted. (https://dms.hiv.health.gov.mw/dataset/covid-19-guidance-for-hiv-services-edition-5/resource/5782288a-dfaf-43d5-8fb8-348c3d11fa33)

2. Introduction and discussion to include mention of global, regional and local context. It does not come out very clearly except if one goes to check references where there is then mention of Africa, CQUIN, Indonesia, South Africa, PEPFAR supported countries etc. May be interesting to know 6-MMD coverage for the study sites at that point in time if information available.

3. Page 7, 1st paragraph states that “However, the impact of scaling-up the 6-MMD strategy has not been evaluated during the COVID-19 pandemic in Malawi. We evaluated the barriers and facilitators of implementing the 6-MMD strategy to better understand its impact during the COVID-19 pandemic among PLHIV receiving care at HIV clinics in Malawi.” It may be useful to further clarify “impact”.

Methods

1. The study population definition statement is inclined towards the sampling or inclusion criteria. I suggest a revision.

2. Is it possible to also have the total number of HCWs who were working in the ART clinics to show how the sample of 77 was decided upon. The selected cadres, what are their roles in the provision of 6-MMD?

3. Are there health care workers who are below the age of 18?

4. Key informant purposive selection: was there any special reason why 100% of clinicians from the survey went on to do the key informant interviews, rather than proportionate apportionment of the cadres? Also, in the survey 10 types of cadres participated but for key informant interviews only 7 types of cadres were included.

5. The framework analysis, sentence number 4: Clarify the attitude of affected patients regarding the policy since patients were not part of the study. Also, the patients part does not appear in the abstract.

6. How many tablets had ODK loaded? How many locations within health facilities had tablet stations for questionnaire filling?

7. Consent is not mentioned for key informant interviews

8. Is there a specific reason why key informant interviews where only done in Chichewa, yet the survey question had English and Chichewa options?

9. Ethical considerations: informed consent mode ? written/verbal

Results:

1. Expert clients: in addition to being HCWs, they double up as patients. Could a sub-analysis of this sub-population have any significant finding?

2. Table 2 results are not included in the discussion or conclusion

3. Were there any significant differences in responses by gender and by cadre?

4. Table 3: Row 12, add (SD) after mean age

5. Table 3: Row 13, mean years worked is presented as a range (1-5years)

6. Table 4: Cotrimoxazole not mentioned, only ART drugs mentioned compared to information on the previous page

7. Table 4: Drugs packed in 1 bottle for 3 months vs a bottle with 6-month supply on previous page. Need to align.

8. Table 4: HCW vs patient factors/barriers/facilitators – classification and presentation

9. Table 4: Orientation and review meeting description: May need additional information for review meetings

10. Table 4: Prior MMD experience: for HCWs or health facilities?

11. Table 4: Improved retention into care: 'The reduced patient visits ensured that patients came on the specified date and improved retention of the patients'. Description may need review

12. Table 4: Delayed or missed patient visits: The assumption is that the appointment are for other services

13. Table 4: Apart from reduced patient visits saving patients money, what were other benefits to patients?

14. May consider including literature on policy for Cotrimoxazole preventive therapy (CPT) for stable patients, Edition 5 of 26th January 2021 also mentions 6MMD policy for ART during stock outs for CPT

15. Page 15, 6th paragraph quotation and page 18: 6 months in 1 bottle, vs 3 months, vs 30 tablets vs type of dolutegravir based regimen vs 90 tablets in a bottle - alignment of HCW knowledge

16. In the methods section the authors state that "The questionnaire had a combination of structured questions and a three-point Likert scale (agree, neutral, disagree), and focused on questions on perceived barriers and facilitators of implementing the 6-MMD strategy". Is table 2 reflective of the structured questions or there are other results/questions for this part?

Discussion:

1. Some themes in Table 4 are evidence of limited HCW knowledge of the policy and could be discussed

2. Drug packaging in fewer bottles vs anonymity may need further explanation for clarity

3. Other potential limitations to consider: patients not included in the study for 1st hand information, study conducted in health facilities with prior 6MMD experience

Conclusion:

1. It is not clear from the analysis/discussion to support the conclusion of understanding of policy as one of the major challenges- is it by frequencies in responses/themes?

Minor Observations

1. 'Please note it is not acceptable for an author to be the sole named individual responsible for ensuring data access' (https://journals.plos.org/globalpublichealth/s/data-availability#loc-acceptable-data-sharing-methods)

2. Missing author affiliation, Stella Mahuva

3. Some punctuation, spacing, section and subsection headings formatting required

4. Unpack CSV abbreviation on 1st use in the document

5. Statistics and data (STATA) software (unpacking abbreviation)

6. ASO abbreviation (unpack for 1st time use)

6. PLOS authors have the option to publish the peer review history of their article (what does this mean?). If published, this will include your full peer review and any attached files.

**Do you want your identity to be public for this peer review?** For information about this choice, including consent withdrawal, please see our Privacy Policy.

Reviewer #1: No

Reviewer #2: No

Reviewer #3: **Yes: **Normusa Musarapasi

---

## [Editor Report · Decision Letter 1]

27 Nov 2024

Barriers and facilitators to implementing six monthly multi-month dispensing of antiretroviral therapy in two urban HIV clinics during the COVID-19 Era in Malawi

PGPH-D-24-01355R1

Dear Dr Chamanga,

We are pleased to inform you that your manuscript 'Barriers and facilitators to implementing six monthly multi-month dispensing of antiretroviral therapy in two urban HIV clinics during the COVID-19 Era in Malawi' has been provisionally accepted for publication in PLOS Global Public Health.

Best regards,

Tsitsi B. Masvawure, Ph.D.

Academic Editor

Thank you for being responsive to the reviewers' comments. I believe that you have sufficiently addressed the concerns raised.